# UNCERTAINTY QUANTIFICATION USING VARIATIONAL INFERENCE FOR BIOMEDICAL IMAGE SEGMENTATION

## ABSTRACT

Deep learning motivated by convolutional neural networks has been highly successful in a range of medical imaging problems like image classification, image segmentation, image synthesis etc. However for validation and interpretability, not only do we need the predictions made by the model but also how confident it is while making those predictions. This is important in safety critical applications for the people to accept it. In this work, we used an encoder decoder architecture based on variational inference techniques for segmenting brain tumour images. We evaluate our work on the publicly available BRATS dataset using Dice Similarity Coefficient (DSC) and Intersection Over Union (IOU) as the evaluation metrics. Our model is able to segment brain tumours while taking into account both aleatoric uncertainty and epistemic uncertainty in a principled bayesian manner.

## 1 INTRODUCTION

Medical image segmentation is a challenging task for medical practitioners. It is costly, takes time and is prone to error. Hence there is a need to automate the manually done segmentation. Lately Neural Networks have shown great potential on a variety of medical image segmentation problems. The challenge with the approaches used in literature is that the model doesn't predict the uncertainty associated. This is where Bayesian methods come into play as it gives a principled way of measuring uncertainty from the model predictions. Measuring uncertainty in the output predictions made by neural networks is important for interpretation and validation. Rather than learning the point estimates, Bayesian Neural Networks (BNN) learns the distribution over the weights. The training process of BNN involves first initializing the parameters of the neural network. Next the weights are sampled from some distribution (like gaussian with zero mean and unit variance) and both the forward pass and backward pass is done to update the weights using the conventional backpropagation algorithm.

Monte Carlo dropout networks (Kingma et al., 2015) use dropout layers to approximate deep Gaussian processes which still lack theoretical understanding. Bayesian Convolutional Neural Network (Gal and Ghahramani, 2015) use variational inference to learn the posterior distribution over the weights given the dataset. The problem with this approach is that it requires a lot of computation involving a lot of parameters, making this technique not scalable in practice. Variational Autoencoder (Kingma et al., 2015) which is based on generative models solves the above problems and has been successful in a number of tasks like generating images, texts, recommender systems etc. This approach comes with several challenges in its own right which have been successfully tackled in the literature. A random variable sampled from posterior distribution has no gradient so the conventional backpropagation techniques can't be applied to it. Local Reparameterization Trick (Kingma et al., 2015) was proposed to tackle this by converting the random variable to a deterministic one for computation. The second challenge was the huge computational requirement since it required weight updates in every iteration. Bayes by Backprop algorithm (Blundell et al., 2015) tackled this by calculating gradients in backpropagation using a scale and shift approach by updating the posterior distribution in the backward pass.

## 2 RELATED WORK

### 2.1 MEDICAL IMAGE SEGMENTATION

The problem of segmenting medical images have been successfully tackled in literature using mainly two techniques, first using a Fully Convolutional Network (FCN) (Long et al., 2015) and second those which are based on U-Net (Ronneberger et al., 2015). The main characteristic of FCN architectures is that it doesn't use fully connected layers at the end which have been used successfully for image classification problems. U-Net methods on the other hand uses an encoder-decoder architecture with pooling layers in the encoder and upsampling layers in the decoder. Skip connections connect the encoder layers to the decoder layer to create an additional path for the flow of gradients back in the backpropagation step. This helps in reducing overfitting due to many parameters involved while training the network.

### 2.2 BAYESIAN NEURAL NETWORK

Lately, there has been a revival of interest in bayesian methods as some of the inherent problems with deep learning could be solved using it. It is a scalable approach of avoiding overfitting in neural networks and at the same time gives us a measure of uncertainty. This is very important in critical applications where not only do we require the predictions made from the model, but also how confident it is while making its predictions. BNN can be considered as an ensemble of neural networks (Gal, 2016). It has two advantages over the standard neural networks, first it avoids overfitting and second it gives a measure of uncertainty involved.

Instead of point estimates, the neural network learns posterior distribution over the weights given the dataset as defined in Equation 1.

$$p(\omega|\mathcal{D}) = \frac{p(\mathcal{D}|\omega)p(\omega)}{p(\mathcal{D})} = \frac{\prod_{i=1}^{N} p\left(y_i|x_i, \omega\right)p(\omega)}{p(\mathcal{D})} \tag{1}$$

The predictive distribution can be calculated by approximating the integral as defined in Equation 2.

$$p\left(y^*|x^*, \mathcal{D}\right) = \int_{\Omega} p\left(y^*|x^*, \omega\right)p(\omega|\mathcal{D})d\omega \tag{2}$$

The challenge is that the posterior is often intractable in nature. To combat this, (Neal, 1993) used Markov Chain Monte Carlo (MCMC) for learning the weights over the bayesian neural networks. Also (Graves, 2011), (Blundell et al., 2015) and (Louizos and Welling, 2016) proposed independently a technique using variational inference techniques for approximating the posterior distribution. KL Divergence between the posterior and the true distribution can be calculated using Equation 3.

$$KL\left\{q_\theta(\omega)\|p(\omega|\mathcal{D})\right\} := \int_{\Omega} q_\theta(\omega)\log\frac{q_\theta(\omega)}{p(\omega|\mathcal{D})}d\omega \tag{3}$$

Alternatively minimizing the KL divergence can be written in another form by maximizing the Evidence Lower Bound (ELBO) which is tractable. This is shown in Equation 4.

$$-\int_{\Omega} q_\theta(\omega)\log p(y|x, \omega)d\omega + KL\left\{q_\theta(\omega)\|p(\omega)\right\} \tag{4}$$

### 2.3 VARIATIONAL INFERENCE

Variational inference finds the parameters of the distribution by maximizing the Evidence Lower Bound. ELBO consists of sum of two terms Kullback-Leibler (KL) divergence between two distributions and the negative log-likelihood (NLL) as defined in Equation 5.

$$\min \mathrm{KL}\left(q_\theta(w)\right)\|p(w|\mathcal{D})) \tag{5}$$

The KL divergence is defined in equation 6.

$$\text{KL}(q(x)) \| p(x)) = -\int q(x) \log \left( \frac{p(x)}{q(x)} \right)$$

(6)

The posterior in the above equation contains an integral which is intractable in nature. The equation can be re written in Equation 7.

$$
\begin{aligned}
\text{KL}\left(q_\theta(w)\right) \| p(w|\mathcal{D})) = \mathbb{E}_{q_\theta(w)} \log \frac{q_\theta(w)p(\mathcal{D})}{p(\mathcal{D}|w)p(w)} &= \\
&= \log p(\mathcal{D}) + \mathbb{E}_{q_\theta(w)} \log \frac{q_\theta(w)}{p(w)} - \mathbb{E}_{q_\theta(w)} \log p(\mathcal{D}|w) \\
&= \log p(\mathcal{D}) - \mathcal{L}(\theta)
\end{aligned}
$$

(7)

The above equation can be decomposed into two parts one of which is the KL divergence between the exact posterior and its variational approximation which needs to be minimized and the second is ELBO term which needs to be maximized. This is shown in Equation 8.

$$\max_\theta \log p(\mathcal{D}) = \max_\theta \left[ \text{KL}\left(q_\theta(w)\right) \| p(w|\mathcal{D})) + \mathcal{L}(\theta) \right]$$

(8)

KL divergence is zero if exact posterior is equal to variational approximation. Since the KL divergence is always greater than zero hence the equation can be approximated by maximizing only the ELBO (Kingma et al., 2015) as defined in equation 9.

$$\mathcal{L}(\theta) = \mathbb{E}_{q_\theta(w)} \log p(\mathcal{D}|w) - \mathbb{E}_{q_\theta(w)} \log \frac{q_\theta(w)}{p(w)} = \mathcal{L}_\mathcal{D} - \text{KL}\left(q_\theta(w) \| p(w)\right)$$

(9)

### 2.4 ALEATORIC UNCERTAINTY AND EPISTEMIC UNCERTAINTY

There are two types of uncertainty - aleatory and epistemic uncertainty where variance is the sum of both these. Bayesian Neural Networks can be considered an ensemble of neural networks initialized randomly which averages the test results in parallel (Gal, 2016). For final predictions, single mean and variance can be estimated as shown in Equation 10 and Equation 11 respectively.

$$\mu_c(x) = \frac{1}{M} \sum_{i=1}^M \hat{\mu}_i(x)$$

(10)

$$\hat{\sigma}_c^2(x) = \frac{1}{M} \sum_{i=1}^M \tilde{\sigma}_i^2(x) + \left[ \frac{1}{M} \sum_{i=1}^M \hat{\mu}_i^2(x) - \hat{\mu}^2(x) \right]$$

(11)

The first term in variance denotes aleatoric uncertainty while the second denotes epistemic uncertainty. Bayesian Neural Network model for uncertainty estimation was done by (Kendall and Gal, 2017) with the last layer representing the mean and variance of logits. The predictive distribution approximating the posterior distribution which gives a measure of uncertainty is defined in Equation 12.

$$q_{\hat{\theta}}\left(y^*|x^*\right) = \int_\Omega p\left(y^*|x^*, \omega\right) q_{\hat{\theta}}(\omega) d\omega$$

(12)

Aleatoric uncertainty is a measure of the variability of the predictions from the dataset hence it is inherent in the data present. The aleatoric uncertainty is the uncertainty that exists inside the dataset. It captures noise inherent in the observations, which arises from the distribution of data. It is highly dependent on bias and distribution inside the input data but not the size of training samples.

Epistemic uncertainty on the other hand is a measure of the variability of predictions from the model which is tied to various metrics used for evaluation like accuracy, loss etc. It represents the uncertainty inside the model. The epistemic uncertainty will decrease with enough training samples. It is resulting from the limitation of knowledge and data of the system.

# 3 METHOD

## 3.1 DATASET

To validate the performance of our proposed approach to generalization, publicly available datasets were used for brain tumour segmentation BRATS18 (Menze et al., 2015) and (Bakas et al., 2018). It contains MRI scans of 175 patients with glioblastoma and lower grade glioblastoma. The images were of resolution $240 \times 240 \times 155$ pixels. The ground truth labels were created by expert neuroradiologists. The sample from the dataset is shown in Figure 2.

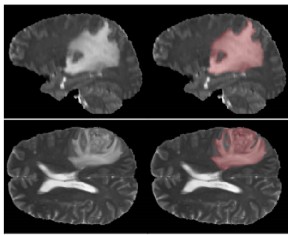

Figure 1: Example of MRI slices and ground truth segmentation

## 3.2 DATA AUGMENTATION

The following data augmentation methods were used to increase the size of dataset:

**1. Rescaling:** We rescale the pixels values by rescaling factor 1/255.

**2. Rotation:** Random rotations with the setup degree range between [0, 360] were used.

**3. Height and Width Shift:** Shifting the input to the left or right and up or down was performed.

**4. Shearing Intensity:** It refers to shear angle (unit in degrees) in counter-clockwise direction.

**5. Brightness:** It uses a brightness shift value from the setup range.

## 3.3 NETWORK ARCHITECTURE

The prior distribution helps to incorporate learning of the weights over the network. Variational Autoencoder has been used successively as a kind of generative model by sampling from the prior distribution in the encoder. The decoder uses the mean vector and standard deviation vector from the latent space to reconstruct the input. Our model uses a similar encoder decoder architecture as that used in VAEs with the input to the encoder coming from a pre trained image segmentation architecture. We tried different backbones which have previously enjoyed success and found original U-Net gave the best results. The input to the encoder only needs the mean vector, the standard deviation vector of the conditional distribution expressing the confidence with which the pixels are correctly predicted. After passing through the encoder, the parameters get converted to a latent representation which is again sampled in a mean and standard deviation vector. The decoder later recovers this back to the original distribution. The conventional backpropagation algorithm is used for training the model with gradient descent. The network architecture is shown in Figure 1.

Let dataset be denoted by $\mathcal{D} : \{(x_i, y_i)\}_{i=1}^{N}$, variational approximation of the posterior distribution by $q_\theta(w)$, encoder as $r_\psi(z|w)$ and decoder as $p_\phi(w)$.

The objective function used in this work is defined in Equation 13:

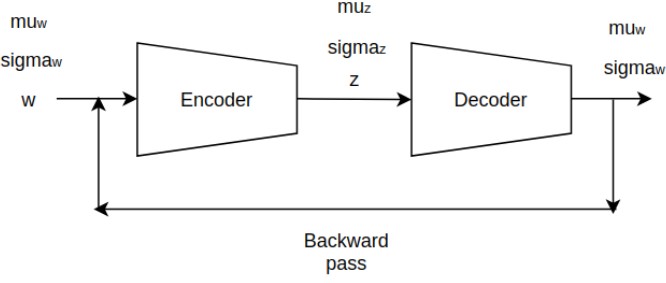

Figure 2: Illustration of our network architecture

$$\mathcal{L}^{\text{approx}} = \sum_i \left[ -\log q_{\theta_i}\left(\widehat{w}^{(i)}\right) - \log r_{\psi^{(i)}}\left(\widehat{z}|\widehat{w}^{(i)}\right) + \log p_{\phi^{(i)}}\left(\widehat{w}^{(i)}|\widehat{z}\right) \right] \quad (13)$$

The network is trained using Gradient Descent as defined in Equation 14 and Equation 15:

$$\theta = \theta + \alpha\nabla_\theta\mathcal{L} \text{ and } \psi = \psi + \beta\nabla_\psi\mathcal{L} \quad (14)$$

$$g_\phi \leftarrow \frac{1}{m}\sum_{k=1}^{m}\nabla_\phi \log p_\phi\left(x^{(k)}|z_\theta\left(x^{(k)}, \epsilon^{(k)}\right)\right) \quad (15)$$

We need as predictions the posterior distribution of the model parameters which is denoted as $q_\theta(w)$.

The complete algorithm used in our work is shown below:

---

**Algorithm 1:** Uncertainty Quantification using Variational Inference for Biomedical Image Segmentation

---

Input: Dataset $\mathcal{D} : \{(x_i, y_i)\}_{i=1}^{N}$
Input: Variational approximation of the posterior distribution $q_\theta(w)$
Input: encoder $r_\psi(z|w)$ and decoder $p_\phi(w)$
**while** *not converged* **do**
  Sample minibatch: $\mathcal{D}^* : \{(x_i, y_i)\}_{i=1}^{M}$
  Sample weights with reparametrization: $\widehat{w}^{(i)} \sim q_{\theta_i}\left(w^{(i)}\right)$
  Sample latent variables with reparametrization: $\widehat{z}^{(i)} \sim r_{\psi^{(i)}}\left(z|\widehat{w}^{(i)}\right)$
  Compute stochastic gradients of the objective:
   $\mathcal{L}^{\text{approx}} = \sum_i \left[ -\log q_{\theta_i}\left(\widehat{w}^{(i)}\right) - \log r_{\psi^{(i)}}\left(\widehat{z}|\widehat{w}^{(i)}\right) + \log p_{\phi^{(i)}}\left(\widehat{w}^{(i)}|\widehat{z}\right) \right]$
  Update parameters $\theta = \theta + \alpha\nabla_\theta\mathcal{L}$ and $\psi = \psi + \beta\nabla_\psi\mathcal{L}$
   $g_\phi \leftarrow \frac{1}{m}\sum_{k=1}^{m}\nabla_\phi \log p_\phi\left(x^{(k)}|z_\theta\left(x^{(k)}, \epsilon^{(k)}\right)\right)$
**end**
Output: $q_\theta(w) -$ posterior distribution of the model parameters

---

### 3.4 EXPERIMENTAL DETAILS

The hyperparameters used in our model are specified in Table 1.

In addition to the above hyperparamaters, cyclical learning rate schedulers and ReduceLROnPlateau was used. In gradient descent, the value of momentum was taken as 0.9, $\gamma$ value of 0.1 and weight decay of 0.0005.

Table 1: Hyperparameters details

| Parameter | Value |
|---|---|
| Batch Size | 16 |
| Optimizer | Adam |
| Learning Rate | 0.001 |
| LR scheduler patience | 10 |
| LR scheduler factor | 0.1 |
| Latent Variable Size | 10 |
| Max epochs | 500 |
| Dropout(Both Training and Test) | 0.3 |

## 3.5 LOSS FUNCTIONS

A combination of binary cross entropy and dice losses have been used to train the network. The first part binary cross entropy is a commonly used loss function for classification problems as shown by (Goodfellow et al., 2016). Every pixel in the image needs to be classified and hence loss function can be written as shown in Equation 18.

$$\mathcal{L}_{CE} = -\sum_{i,j} y_{i,j} \log \widehat{y}_{i,j} + (1 - y_{i,j}) \log (1 - \widehat{y}_{i,j}) \quad (16)$$

The problem with binary cross entropy loss is that it doesn't take into account the class imbalance as the background is the dominant class. This is one of fundamental challenges in semantic segmentation problems. Dice Loss is robust to the aforementioned problem which is based on Dice Similarity Coefficient as defined in Equation 19.

$$\mathcal{L}_{DICE} = \sum_{i=1}^{N} \frac{FN_i + FP_i}{2TP_i + FN_i + FP_i} = \sum_{i=1}^{N} \left(1 - DSC^{(i)}\right) \quad (17)$$

Both the loss terms were combined in a single term with more weight given to the Dice Loss term since it handles the class imbalance problem better. This is shown in Equation 20.

$$\mathcal{L} = 0.9 \cdot \mathcal{L}_{DICE} + 0.1 \cdot \mathcal{L}_{CE} \quad (18)$$

## 3.6 EVALUATION METRICS

Evaluation metrics for semantic segmentation problems which have often been used in literature are Dice Similarity Coefficient (DSC) also known as F1-score and Intersection over union (IoU). The corresponding equations are shown in Equation 16 and Equation 17 respectively.

$$\text{DSC} = \frac{2TP}{2TP + FN + FP} \quad (19)$$

$$\text{IoU} = \frac{TP}{TP + FN + FP} \quad (20)$$

True positive (TR), false negative (FN) and false positive (FP) number of pixels is calculated separately for each image and averaged over the test set. The ground truth is labelled manually by experts which are compared against.

## 4 RESULTS

Figure 3 displays how accuracy (y-axis) behaves based on how much the data (x-axis) holds.

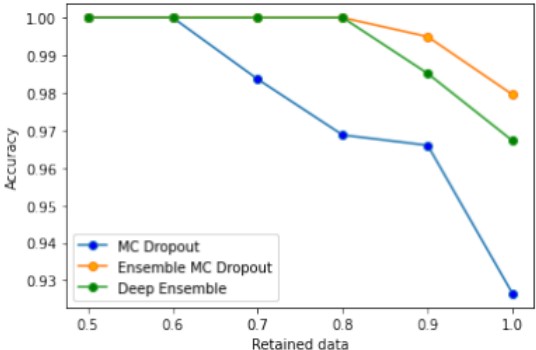

Figure 3: Accuracy vs data distribution of our network using uncertainty quantification approach on validation/test sets.

We have to generate the uncertainty value of each input based on the predicted output. This can be done using the below uncertainty quantification methods:

**1. MC dropout:** This is a method based on Bayesian Neural Networks and activated dropout function at inference time. This can be defined using the below equation:

$$\text{Uncertainty}_{mcd} = -\left( \sum_{i=1}^{M} \sum_{j=1}^{c} p_j \log p_j \right) / M_i \tag{21}$$

where $M$ is MC sample size, $c$ is the number of classes, $i$, $j$ are the indexes of $M$, $c$ and $p_i$ is the probability distribution of each image.

**2. Deep ensembles:** We collect the results of each image trained on five models and compute the average mean. We then calculate the entropy of these mean values as the uncertainty value on each image.

**3. Ensemble MC dropout:** This method is a combination of MC dropout and deep ensembles. It contains 5 Bayesian Neural Networks and thus following a similar way to compute MC dropout but with the average value of these five models. This can be defined using the below equation:

$$Uncertainty_{ens-mcd} = -\sum_{k=1}^{D} ((\sum_{i=1}^{M} \sum_{j=1}^{c} p_j \log p_j)/M_i \, /D_k \tag{22}$$

where $D$ presents how many models are there in ensemble MC dropout, $k$, $i$, $j$ are the indexes of $D$, $M$, $c$ and other variables same as MC dropout uncertainty formula above.

Table 2: Uncertainty mean value of samples in the validation/test sets by our network.

| Model | Uncertainty(Val) | Uncertainty(Test) |
|---|---|---|
| MC dropout | 0.1607 | 0.1217 |
| Deep ensembles | 0.1204 | 0.1146 |
| Ensemble MC dropout | 0.2235 | 0.1213 |

The Mean Dice Similarity value for various backbone architectures compared against different train size values are shown in Table 2.

The IOU value for various backbone architectures compared against different train size values are shown in Table 3.

The predicted segmentation along with uncertainty involved in segmentation is shown in Figure 4:

Table 3: Mean Dice Similarity metrics for the experiments

| Train Size | DSC |
| --- | --- |
| 5 | 53.1 |
| 10 | 56.6 |
| 15 | 60.8 |
| 20 | 64.3 |

Table 4: Intersection over Union metrics for the experiments

| Train Size | IOU |
| --- | --- |
| 5 | 48.4 |
| 10 | 50.6 |
| 15 | 53.1 |
| 20 | 55.8 |

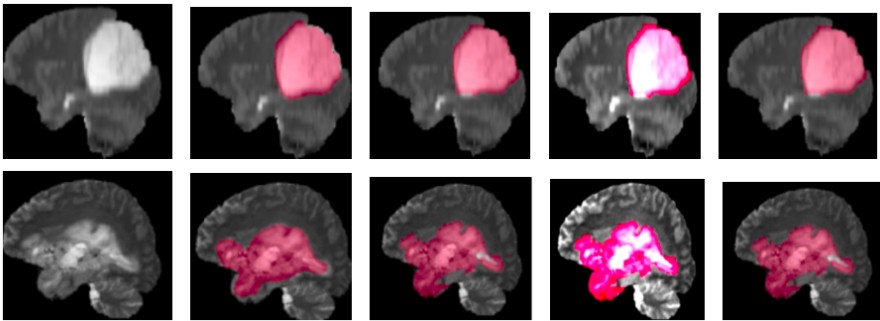

Figure 4: Examples of models predictions on test samples, compared to ground truth segmentation. First column: input image, second column: ground truth segmentation, third column: predicted segmentation, fourth column: aleatoric uncertainty and fifth column: epistemic uncertainty.

The darker(dull) color denotes more confidence while the lighter(brighter) color means the model is less confident while segmenting those areas. Another point to be noted is that aleatoric uncertainty and epistemic uncertainty are co-related i.e with the increase in confidence taking into account aleatoric uncertainty while segmenting brain tumours results in increase of epistemic uncertainty and vice versa.

## 5 CONCLUSIONS

In this work, we proposed a way to quantify uncertainty in the context of medical image segmentation. Our network is based on an encoder decoder framework similar to that used by VAEs. The weights of the network represent distributions instead of point estimates and thus give a principled way of measuring uncertainty at the same time while making the predictions. Our model uses bayesian neural networks for both the encoder and decoder. The inputs to encoder come from pre trained backbones like U-Net, V-Net and FCN sampled from conditional distribution representing the confidence with which pixels are labelled correctly. We evaluated our network on publicly available BRATS dataset with our model outperforming previous state of the art using DSC and IOU evaluation metrics.

ACKNOWLEDGMENTS

We would also like to thank Nvidia for providing the GPUs.

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
