# OpenReview forum: "UNCERTAINTY QUANTIFICATION USING VARIATIONAL INFERENCE FOR BIOMEDICAL IMAGE SEGMENTATION"
_ICLR.cc/2022/Conference — ICLR 2022 Submitted_

### Official Review · Reviewer_MfyG · 2021-10-24

**Correctness:** 4
**Technical Novelty And Significance:** 1
**Empirical Novelty And Significance:** 1
**Recommendation:** 1
**Confidence:** 5

**Main Review:**

Originality: The novelty is quite limited, as VAE has been explored in the medical domain [1]. Plus, it is a common practice in medical image segmentation to estimate the uncertainty of segmentation result by performing multiple times of dropout at test time [2]. There is not any improvement on their VAE architecture as well.

Clarity: The paper is clearly written and easy to follow, but has some typos.

Overall, the novelty of the paper does not meet the standard of ICLR, and there is not any meaningful contribution in this paper. So I vote for rejection.

[1] Sedai, Suman, et al. "Semi-supervised segmentation of optic cup in retinal fundus images using variational autoencoder." International Conference on Medical Image Computing and Computer-Assisted Intervention

[2] Wang, Guotai, et al. "Aleatoric uncertainty estimation with test-time augmentation for medical image segmentation with convolutional neural networks." Neurocomputing 338 (2019): 34-45.


**Summary Of The Paper:**

This paper propose to use VAE to perform the brain tumor segmentation task and uncertainty estimation.

**Summary Of The Review:**

The paper does not have any meaningful contribution.

---

### Official Review · Reviewer_5veL · 2021-10-31

**Correctness:** 2
**Technical Novelty And Significance:** 1
**Empirical Novelty And Significance:** 1
**Recommendation:** 1
**Confidence:** 4

**Main Review:**

The paper is a number of weaknesses:

- No attempt is made to put the work in the context of related work. There many, many works on uncertainty estimation in the context of biomedical image segmentation (see below). The authors give the impression that they are the first to tackle this challenge.

Stochastic Segmentation Networks, Monteiro et al, NeurIPS 2020
A probabilistic U-net for segmentation of ambiguous images, Kohl et al, NeurIPS 2018
many more

- The paper has no contribution section where the contributions of the paper are explicitly listed. From the description of the methodology segmentation it seems that that there no novel contributions, all of the components of the method are well known (augmentation, loss function, variational inference, etc).

- The results look rather underwhelming, no attempt is made at comparing the results to those of other methods that have been evaluated on the BRATS challenge.

- No attempt is made at evaluating the usefulness of the uncertainty estimates for the users.


**Summary Of The Paper:**

The paper propose a deep learning based method for taking into account both aleatoric uncertainty and epistemic uncertainty in biomedical image segmentation tasks. The proposed method is based on variational inference techniques with a standard encoder-decoder CNN architecture. The method is applied to brain tumour MR images from the standard BRATS segmentation challenge.

**Summary Of The Review:**

The paper makes no significant technical contribution or innovation and does not offer any strong results.

---

### Official Review · Reviewer_UXjb · 2021-11-01

**Correctness:** 1
**Technical Novelty And Significance:** 1
**Empirical Novelty And Significance:** 1
**Recommendation:** 1
**Confidence:** 5

**Main Review:**

There are multiple major issues about the paper.

- Methods description is very difficult to understand. It is not very clear to me how the proposed method works
- Experimental evaluations are quite limited. There are some Dice score and IoU values, but there is no comparison with another method in the literature.

**Summary Of The Paper:**

The paper proposes a method for uncertainty quantification for biomedical image segmentation. The proposed method takes mean and std of segmentation from a backbone segmentation model and trains a VAE on top of it. The method is evaluated on only BRATS dataset.

**Summary Of The Review:**

There are multiple major issues about the paper.

- Methods description is very difficult to understand. It is not very clear to me how the proposed method works
- Experimental evaluations are quite limited. There are some Dice score and IoU values, but there is no comparison with another method in the literature.

---

### Official Review · Reviewer_YKhZ · 2021-11-02

**Correctness:** 3
**Technical Novelty And Significance:** 1
**Empirical Novelty And Significance:** 2
**Recommendation:** 1
**Confidence:** 3

**Main Review:**

Strengths:

-- U-Net architecture as backbone is a strong choice for the medical image segmentation and the proposed method was evaluated on publicly available dataset.

-- The paper provides results of 3 different uncertainty quantification methods ("MC dropout", "Deep Ensembles", "Ensemble MC dropout"), , and DSC/IOU of varying train sizes.

-- The paper includes background of the existing theory and sentences are usually clear.


Weaknesses:

-- The novelty of the paper is quite limited. The authors are encouraged to re-evaluate the study from the methodological and/or clinical perspective, and provide clear and specific points about their novel contributions at the end of the introduction. Related work section does not include any information about the existing related studies in the literature (e.g. [1, 2, 3]).

-- The paper could possibly be considered as an application paper but the contributions in the paper have marginally low significance because the paper provides very limited empirical insights with already available methods with slight changes.

-- The organization of the paper should be improved. For example, the majority of the paper contains just definitions of the known formulas. Most equations presented in the paper are missing references. Can the authors clarify whether there are any novel equations in the paper? Also, some methodological details are missing. The authors stated "Our model uses a similar encoder decoder architecture as that used in VAEs with the input to the encoder coming from a pre trained image segmentation architecture.". What type of VAE architecture was this? Were there any modifications? With which data was the encoder pre-trained? The authors are encouraged to add these details to the paper.

-- I also highly recommend that the authors compare their methodology with the previous works (e.g. [1]). In its current form, the paper does not provide any comparison with the existing studies in the literature, or a comparison of different backbones even though the authors stated that "We tried different backbones which have previously enjoyed success and found original U-Net gave the best results." in the paper. Can the authors clarify what those backbones are? I assumed these are U-Net, V-Net and FCN as stated in the conclusion (without any references) but the authors are encouraged to list them in the method section (with references to the specific architectures unless they are novel).



[1] Kwon, Yongchan, Joong-Ho Won, Beom Joon Kim, and Myunghee Cho Paik. "Uncertainty quantification using Bayesian neural networks in classification: Application to biomedical image segmentation." Computational Statistics & Data Analysis 142 (2020): 106816.

[2] Nair, Tanya, Doina Precup, Douglas L. Arnold, and Tal Arbel. "Exploring uncertainty measures in deep networks for multiple sclerosis lesion detection and segmentation." Medical image analysis 59 (2020): 101557.

[3] Ghoshal, Biraja, Allan Tucker, Bal Sanghera, and Wai Lup Wong. "Estimating uncertainty in deep learning for reporting confidence to clinicians in medical image segmentation and diseases detection." Computational Intelligence 37, no. 2 (2021): 701-734.

**Summary Of The Paper:**

The paper proposes a method to quantify uncertainty in medical imaging, which is an important task for clinical applications, with variational inference. It uses the U-Net architecture and BRATS18 dataset for evaluation. It quantifies uncertainty with 3 methods and evaluates the predictions with Dice score (DSC) and Intersection over Union (IOU).

**Summary Of The Review:**

The novelty of the paper is quite limited, and the paper should be rewritten to include important metholodogical details as well as comparisons with the existing studies to assess its impact and significance.

---

### Decision · Program_Chairs · 2022-01-20

**Decision:**

Reject

**Comment:**

The paper introduces a method for uncertainty quantification for medical applications, which quantifies both aleatoric and epistemic components.

The paper initially received three strong reject recommendations. The main limitations pointed out by reviewers relate to the limited contributions (either methodological or applicative and clinical), the lack of positioning with respect to related works, the presentation needing improvement and the lack of experimental comparison with respect to recent relevant baselines.
No rebuttal was provided. \
The AC carefully read the submission and agrees that the paper is premature for publication in the current form. Therefore, the AC recommends rejection.